# Botulinum Toxin Therapy for Spasmodic Dysphonia in Japan: The History and an Update

**DOI:** 10.3390/toxins14070451

**Published:** 2022-07-01

**Authors:** Masamitsu Hyodo, Kahori Hirose, Asuka Nagao, Maya Nakahira, Taisuke Kobayashi

**Affiliations:** 1Department of Otolaryngology-Head and Neck Surgery, Kochi Medical School Hospital, Kohasu, Okou-cho, Nankoku 783-8505, Japan; hiroseka@kochi-u.ac.jp (K.H.); nagaoa@kochi-u.ac.jp (A.N.); tkobayashi@kochi-u.ac.jp (T.K.); 2Rehabilitation Department, Kochi Medical School Hospital, Kohasu, Okou-cho, Nankoku 783-8505, Japan; jm-nakahira-m@kochi-u.ac.jp

**Keywords:** spasmodic dysphonia, nationwide survey, placebo-controlled double-blind clinical trial, type 2 thyroplasty using titanium bridge

## Abstract

Spasmodic dysphonia (SD) is a rare neurological disorder that impairs phonatory function by triggering involuntary and intermittent contractions of the intrinsic laryngeal muscles. SD is classified into three types: adductor SD (AdSD), abductor SD (AbSD), and mixed SD. Of these, AdSD accounts for 90–95% of disease; younger females are predominantly affected. Botulinum toxin injection into the laryngeal muscles is safe, minimally invasive, and very effective. Here, we review the history of clinical research for SD conducted in Japan. The first use of botulinum toxin injection therapy to treat SD in Japan was by Kobayashi et al. in 1989. The group developed an objective mora (syllable) method to evaluate SD severity. Recently, we conducted a placebo-controlled, randomized, double-blinded clinical trial of botulinum toxin therapy for AdSD and an open-label trial for AbSD to obtain the approval of such therapy by the Japanese medical insurance system. The mora method revealed significant voice improvement and the evidence was of high quality. Additionally, a clinical trial of type 2 thyroplasty using titanium bridges confirmed the efficacy and safety of such therapy. These studies broadened the SD treatment options and have significantly benefited patients.

## 1. Introduction

Spasmodic dysphonia (SD) is a rare neurological voice disorder featuring involuntary and intermittent contractions of the laryngeal muscles [1]. Depending on the affected muscles, SD is classified into adductor, abductor, and mixed types. Adductor SD (AdSD) is characterized by strained, strangled, and effortful vocalization caused by intermittent hyperadduction of the true (and often also the false) vocal folds, blocking phonatory airflow [1,2]. Abductor SD (AbSD) is characterized by intermittent breathiness, pitch alterations, and aphonia because of spasmodic glottal widening during phonation. Mixed SD exhibits the features of both of the above conditions. SD is considered to be a focal dystonia that may accompany other dystonic diseases. The etiology of SD remains controversial; however, recent studies have suggested that the combination of a genetic factor with environmental modifiers triggers disease development [3]. Three physiological mechanisms may be involved: decreased inhibition, increased plasticity, and abnormal sensory input to the central nervous system on voice production. The emerging picture of SD is that of a disordered inhibition in response to sensory feedback during phonation [3,4]. The treatment modalities include voice therapy, botulinum toxin injection, and surgical interventions. Voice therapy seeks to reduce hypertonic glottal closure and includes humming, speaking slowly, volume control, exhalation before speaking, relaxation, and environmental adjustment [1]. Although this improves the voice with minimal effort, voice therapy is not curative, and the therapeutic effects are limited [5]. Surgical approaches for AdSD aim to prevent hyperadduction of the vocal folds, and include unilateral recurrent laryngeal nerve sectioning [6,7], thyroarytenoid myectomy [8,9], selective adductor denervation-reinnervation surgery [10], and type 2 thyroplasty [11,12]. These procedures reduce strangled and effortful vocalization. However, hoarseness may develop postoperatively, and the long-term results remain unclear. Moreover, there is no effective surgical intervention for AbSD.

Botulinum toxin chemically denervates an affected muscle by blocking acetylcholine release at the presynaptic motor nerve terminal. Thus, acetylcholine does not reach the postsynaptic motor endplate, preventing nerve transduction [13]. Selective injection of botulinum toxin into the laryngeal muscles addresses the laryngeal pathology via selective chemodenervation. The therapy is less invasive than surgery, and very effective [14,15]. Botulinum toxin is today the SD treatment of choice worldwide [16]. Here, we review the history of botulinum toxin therapy for SD both globally and in Japan, and we describe the limitations.

## 2. Epidemiological Surveys

SD incidence and prevalence have been epidemiologically surveyed. Nutt et al. [17] conducted a population-based epidemiological study in Rochester, NY, USA, in 1988; the SD prevalence was 5.2 per 100,000 (95% confidence interval [CI] 1.1–15.1). Duffey et al. [18] in 1998 and the Epidemiological Study of Dystonia in Europe Collaborative Group [19] in 2000 reported lower prevalences of 0.8 (0.5–1.3) in northern England and 0.7 (0.5–0.9) in eight European countries. Konkiewitz et al. [20] performed an epidemiological study of dystonia in Munich, Germany; the SD prevalence was 1.0/100,000 (0.4–1.5). Asgeirsson et al. [21] reported that the prevalence of primary laryngeal dystonia was 5.9/100,000 in Iceland. The National Spasmodic Dystonia Association of the USA estimated that 35,000–50,000 North Americans suffer from SD [22]; thus, 9.7–13.8/100,000. In Japan, Yamazaki [23] conducted a questionnaire survey of 81 university hospitals in 2001. There were 224 SD patients (90.5% with AdSD and 9.5% with AbSD). The age at onset ranged from 14 to 77 years (mean 36.7 years). The male-to-female ratio was 1:4.4. The SD prevalence was estimated to be 0.94/100,000. We also performed a nationwide survey of SD in Japan in 2015 [24]. Of all SD patients, 93.2% had AdSD, 5.7% AbSD, and 1.0% mixed SD. The mean age was 38.9 years and over half (59.0%) were in their 20s and 30s. Females predominated; the male-to-female ratio was 1:4.1. Both surveys evaluated treatment modalities. Yamazaki [23] found that about 27% of AdSD patients were treated with botulinum toxin and 6% surgically (Figure 1). In contrast, our 2015 study revealed significant increases in both treatments (to 41 and 25%, respectively) [24]. This may be because both medical professionals and patients had become more familiar with the effectiveness of such treatments. In 2003, Inoue et al. [25] surveyed the members of the Japanese SD Patients’ Association. The most frequent treatment was botulinum toxin injection (88%) followed by voice therapy (60%), surgery (33%), and medication (19%). However, botulinum toxin therapy and surgery were performed in only a few hospitals and clinics. In other words, no standardized treatment was implemented by most Japanese medical institutions.

## 3. Diagnosis of SD

Basically, diagnosis depends on the experience of each clinician by hearing the patient’s voice characterized by intermittent strained and effortful phonation manner for AdSD and intermittent breathy hoarseness and aphonia for AbSD. However, SD is rare and most of the clinicians have little experience of seeing SD patients. Therefore, its diagnosis is difficult and often delayed [24,26]. We thus proposed diagnostic criteria based on the clinical characteristics [24,27]. They consist of the main and accompanying symptoms, clinical findings during phonation, the treatment response, and the differential diagnoses. The differential diagnoses include voice tremor, muscle tension dysphonia, psychogenic dysphonia, and stuttering. Definite and possible SD are diagnosed depending on the number of items that meet the criteria. This diagnostic criterion has not been confirmed and its validity is currently being verified. Ludlow et al. [24] also proposed a diagnostic tool with a three-step procedure that consists of a screening questionnaire (possible SD), clinical speech examination (probable SD), and laryngeal examination (definite SD) [26]. The procedure is concise and clinically useful. These tools help clinicians to diagnose SD early and accurately.

## 4. Botulinum Toxin Therapy

### 4.1. Historical Review of Botulinum Toxin Therapy

Laryngeal injection of botulinum toxin to treat SD was first performed in 1984 by Blitzer et al. [28]. Thereafter, hundreds of articles were published; most of them found it to be effective [14,29,30,31]. Blitzer et al. [14] treated 1300 AdSD patients over 24 years and reported a 91.2% response rate. The average duration of the benefit was 15.1 weeks. Tisch et al. [27] treated 144 patients (1093 injections); 81.9% evidenced excellent or very good outcomes. Novakovic et al. [30] treated 133 patients (1457 injections). The Voice Handicap Index (VHI) (the patient’s perception of the impact of the voice disorder) improved from 22.3 pre-injection to 12.7 post-injection. Botulinum toxin therapy is generally safe; only transient hoarseness and liquid aspiration have been reported. The American Academy of Otolaryngology—Head and Neck Surgery recommend botulinum toxin as the treatment of choice for AdSD (Clinical Practice Guideline: Hoarseness [Dysphonia]) [32]. It also treats AbSD, but less effectively due to the technical difficulty in approaching the target muscle, the posterior cricopharyngeal muscle [33,34].

In the previous literature, botulinum toxin type A has been primarily used to treat SD and type B is much less used. Blitzer compared treatment effect and safety between type A toxin (Botox) and type B toxin (Myobloc) [35]. They showed that the onset of action of type B was more rapid with a shorter duration of benefit. The safety profile for A and B toxins appeared the same. They showed a conversion factor of 52.3:1 Myobloc to Botox. In Japan, only type A toxin has been used for SD.

### 4.2. Procedure of Botulinum Toxin Therapy

Injections of botulinum toxin have emerged as the preeminent and favorable treatment approach due to its high efficacy rate and less invasiveness [1,28]. It is directly injected into the thyroarytenoid (TA) muscle either percutaneously, transorally, or transnasally in AdSD. The transcutaneous approach is via the cricothyroid membrane using an electromyography (EMG) recording needle to locate the TA muscle, and this is the most common procedure in Japan. As an injection needle, we use Teflon-coated 26G monopolar needle (Ambu Neuroline Inoject). Transoral injection under endoscopic observation and transnasal transendoscopic injection are also alternative approaches. In AbSD, the target muscle is the posterior cricoarytenoid muscle and there are two approaches to inject: a percutaneous posterior-lateral approach and a transnasal endoscopic approach. A percutaneous injection under EMG monitoring is commonly used in Japan.

There is no optimal dosage of BT, as it varies greatly depending on the severity and responsiveness of the patient, but in Japan, the standard dose of botulinum toxin type A (Botox) is 1 to 2.5 U for AdSD and 5 U for AbSD. The dosage can be changed depending on the response after administration, and it has been reported that the dosage often decreases gradually with repeat injections [36].

Botulinum toxin has been injected unilaterally or bilaretally in AdSD. A meta-analysis of the literature suggests that neither unilateral nor bilateral injection is consistently associated with outcomes [1,37,38], although some of the literature concluded that unilateral injection resulted in better functional results [1,39]. In AbSD, simultaneous bilateral injection is prohibited by the Pharmaceuticals and Medical Devices Agency Japan due to a potential risk of dyspnea due to the impairment of the glottal opening, although its safety has been supported by the literature [40]. Contra-indication of botulinum toxin therapy includes a history of hypersensitivity to this drug, coexistence of neuromuscular junction diseases, moderate to severe swallowing disorders, or vocal fold paralysis.

### 4.3. Botulinum Toxin Therapy in Japan

#### 4.3.1. Contributions of the Kobayashi Group

In Japan, botulinum toxin therapy for SD was first performed by Kobayashi et al. in 1989 [41]; that group then led the way in terms of reporting the effectiveness [42]. Dysport was initially injected under EMG monitoring. They had four options: 2.5 and 5 U unilaterally or bilaterally. The optimal option was determined both objectively and subjectively after repeated injections. Of all patients, 38 and 2% received unilateral injections of 2.5 and 5 U, respectively, and 16 and 9% bilateral injections of 2.5 and 5.0 U. The average period of effectiveness was 17.3 weeks. Recently, the group reported the longitudinal changes in the effectiveness after Botox injection [43] (Figure 2). Commonly, subjective voice improvement was apparent within 1.8 ± 0.8 days, and it decreased as breathy hoarseness commenced on the following few days. As the hoarseness gradually resolved, the voice subjectively improved to reach the best from 2.9 ± 2.0 to 6.3 ± 2.9 weeks. The subjective voice score gradually deteriorated after 8.6 ± 2.8 weeks to lead to the next therapy. The mean interval between the therapies was 3.1 ± 0.7 months. The group developed the objective mora method to assess treatment effectiveness [44,45]. A mora is the minimal sound unit of Japanese words (corresponding to an English syllable). The phonatory disorders of SD are associated with disordered mora production. SD is assessed objectively and quantitatively by counting the number of aberrant morae apparent when a patient reads a standardized sentence aloud, reflecting the severity of dysphonia. As described below, this method was used in a Japanese clinical trial of botulinum toxin therapy [46,47].

However, the group identified certain problems with the therapy. First, botulinum toxin therapy was performed at only a few hospitals. SD patients had to spend money and time to receive treatment. Second, Botox and Dysport had not been approved for use in SD therapies; thus, the costs were not recoverable from medical insurers. Both clinicians and patients desired the formal approval of botulinum toxin therapy in Japan.

#### 4.3.2. Randomized, Placebo-Controlled, Double-Blinded Clinical Trials of Botulinum Toxin Therapy

In 2017, the use of botulinum toxin to treat SD had been approved in only 10 countries (Australia, Mexico, Peru, Chile, Colombia, Ecuador, Honduras, Guatemala, Bolivia, and Nicaragua). It had not been approved in any Asian, North American, or European country. Thus, we performed a multi-center, randomized, placebo-controlled, double-blinded parallel-group comparison/open-label clinical trial of botulinum toxin type A (Botox) therapy for AdSD and an open-label trial for AbSD in Japan [46,47]. The study was an investigator-initiated clinical trial in accordance with Good Clinical Practice guidelines. The primary endpoint was a change in the number of disordered morae 4 weeks after botulinum toxin injection. Aberrant morae were evaluated when AdSD and AbSD patients vocalized standardized sentences with 25 and 27 morae, respectively. The secondary endpoints included changes in objective parameters (the number of aberrant morae, the score on the Grade, Roughness, Breathiness, Asthenia, and Strain [GRBAS] scale; perceptual evaluation of voice quality) and subjective parameters (the Voice Handicap Index (VHI]) score, a visual analog scale (VAS) score) over the entire study period.

The results were reported [46,47]. In total, 22 AdSD patients (11 in the Botox group and 11 in the placebo group) and 2 AbSD patients were enrolled. In the AdSD patients, the changes in the aberrant mora numbers 4 weeks after injection were −7.0 ± 2.30 and −0.2 ± 0.46 in the Botox and placebo groups, respectively. The least mean-squares difference between the two groups was significant. Compared to the baseline, the numbers of aberrant morae decreased significantly 2–12 weeks post-injection, with a peak at 2 weeks (Figure 3). The strain element (S) of the GRBAS scale also decreased from 2 to 8 weeks. The VHI score decreased significantly from the baseline over 4 to 12 weeks but did not show any significant change at 2 weeks (Figure 4). The VAS score revealed some post-injection improvement, but the difference was not significant. Improvements in subjective parameters were noted about 2 weeks after the objective parameters improved, attributable to the transient breathy hoarseness that develops after treatment. Although there was no statistical significance, the change values in all parameters tended to be greater in the younger group (<40 years) than in the older group (≥40 years). We considered that younger patients have more plasticity of the disordered central nervous system network [47]. There was no difference in therapeutic response by sex, and the same was reported in the previous literature [47]. On the other hand, Lerner et al. [48] reported that female patients needed a higher botulinum toxin dose for symptom control than males. They speculated that this result was explained by a possible inverse relationship between optimal botulinum toxin dose and vocal fold mass and possibly greater neutralizing antibody formation among female patients. The most common adverse event was breathy voice disorder (77.3%) followed by aspiration when drinking liquid (40.9%). Both were mild and resolved by 25.8 and 15.8 days on average. For AbSD patients, the open-label study showed that the numbers of aberrant morae tended to decrease after Botox injection.

Thus, we concluded that botulinum toxin therapy was efficacious and safe for both AdSD and AbSD patients. We also proposed diagnostic criteria; these are currently being validated. With these works, Botox therapy for SD was formally approved in Japan in 2018 [27,46]. It is today available at more than 60 hospitals and clinics all over the country, greatly benefiting patients.

## 5. Surgical Treatments as an Alternative Therapy

The disadvantages of botulinum toxin therapy include the need for repeat injections, the post-injection hoarse voice, and the cost of long-term treatment. Surgical interventions are chosen by some patients who have difficulty continuing botulinum toxin therapy for these reasons. In Japan, thyroarytenoid myectomy commenced in the early 2000s [9,49]. Isshiki [11] reported the utility of type 2 thyroplasty in 2001. Sanuki et al. in their group reported that use of a titanium bridge to fix dilated thyroid cartilage improved vocal symptoms long-term [12,50]. However, the titanium bridge had not yet been approved as a surgical device. Thus, they conducted an investigator-initiated clinical trial that demonstrated efficacy and safety [51]. With their work, the titanium bridge was approved in 2017 and medical insurance reimbursements commenced in 2018 in Japan. The procedure is becoming popular in Japan and will expand worldwide.

## 6. Role of Botulinum Toxin Therapy

Botulinum toxin therapy is less invasive than surgical intervention and better than voice therapy. It is the preferred first-line treatment of the Clinical Practice Guideline: Hoarseness (Dysphonia) of the USA [32] and the Clinical Practice Guideline for the Diagnosis and Management of Voice Disorders of Japan [52]. Figure 5 shows our treatment algorithm [27]. First, voice therapy should seek to reduce secondarily acquired effortful phonation. This improves the outcomes of botulinum toxin therapy and facilitates a differential diagnosis of SD from muscle tension or psychogenic dysphonia. Next, botulinum toxin is injected regardless of SD severity. However, this is not curative; repeat injections are usually needed long-term. Thus, surgery may be indicated for patients with moderate-to-severe AdSD; the effects are long-term. If symptoms recur after surgery, supplementary botulinum toxin injection is possible. There is no effective surgical treatment for AbSD; botulinum toxin injection is the only option.

## 7. Limitations and Significance of This Review

In this paper, we reviewed a history of the clinical research for SD in Japan, particularly focusing on botulinum toxin therapy. This review is not a meta-analysis or systematic review. Moreover, the history of clinical studies in countries other than Japan and details of the evidence or the long-term course of the treatments have not necessarily been reviewed enough. However, much research on SD has been conducted in Japan which includes epidemiological surveys, proposals of diagnostic criteria, establishment of the mora method, and clinical trials of botulinum toxin (Botox) and thyroplasty using a titanium bridge (Table 1) [23,24,27,44,46,51]. Although there are some issues which need further study, they have greatly contributed to the improvement of knowledge and the ability to choose the appropriate treatment for clinicians treating SD patients. We hope that these studies will promote further research around the world.

In Japan, much research has been conducted through epidemiological surveys, proposals of diagnostic criteria, the establishment of the mora method, and clinical trials of botulinum toxin (Botox) and thyroplasty using a titanium bridge. Although these achievements have greatly contributed to the clinical practice of SD, there still are some issues to be investigated.

## 8. Conclusions

SD challenges neurologists and laryngologists in terms of its etiology, diagnosis, and treatment. However, recent research has improved our knowledge. Japanese scholars have made some important contributions, especially in terms of providing high quality evidence of the effectiveness and safety of botulinum toxin treatment and the development of type 2 thyroplasty using a titanium bridge. Further work on SD etiology and treatment is currently being conducted worldwide.

## Figures and Tables

**Figure 1 toxins-14-00451-f001:**
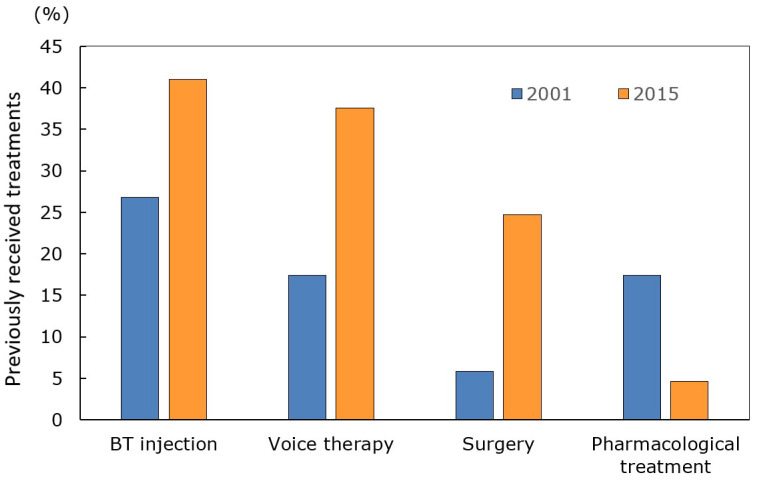
Treatment modalities the patients had previously received. In 2001, 27% of patients had received botulinum toxin injection and 17% completed voice therapy or pharmacological treatment. In 2015, botulinum toxin injection, voice therapy, and surgery became more common. BT: botulinum toxin.

**Figure 2 toxins-14-00451-f002:**
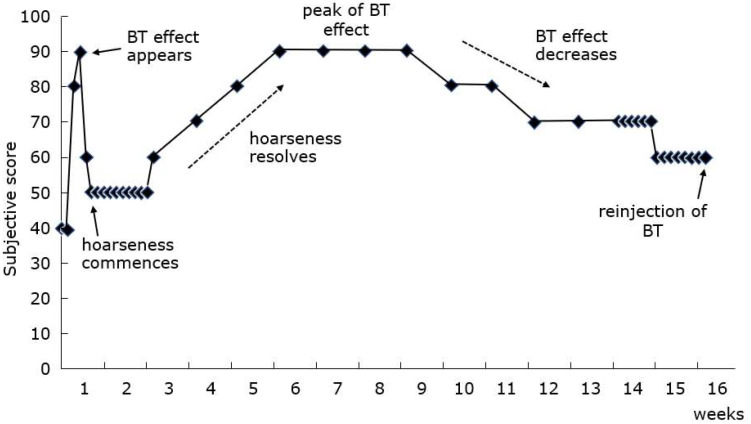
Common longitudinal changes in subjective scores after BT injection. (Reproduced with permission from Ref. [43] 2017, published by The Japan Society of Logopedics and Phoniatrics). Treatment effect is apparent on day 2, but it decreases as hoarseness commences. With a gradual resolve of hoarseness, subjective voice improvements are noted; the improvements peak from 5 to 8 weeks. Thereafter, the subjective score deteriorates by 15 weeks to lead to the next therapy. Y-axis indicates self-assessment score (100: no impairment of voice, 0: possible worst). BT: botulinum toxin.

**Figure 3 toxins-14-00451-f003:**
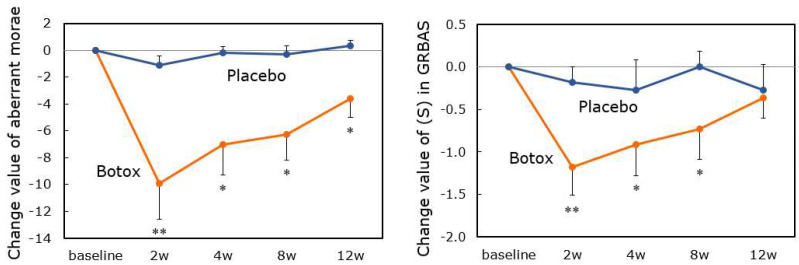
Change values of objective parameters after Botox/placebo injection (Mean ± SE). The number of aberrant morae and the (S) score of the GRBAS scale decrease significantly 2−12 and 2−8 weeks, respectively, after Botox injection, with peaks at 2 weeks. (*: *p* < 0.05, **: *p* < 0.01; vs. baseline) This means Botox injection improves the voice objectively after the 2nd week.

**Figure 4 toxins-14-00451-f004:**
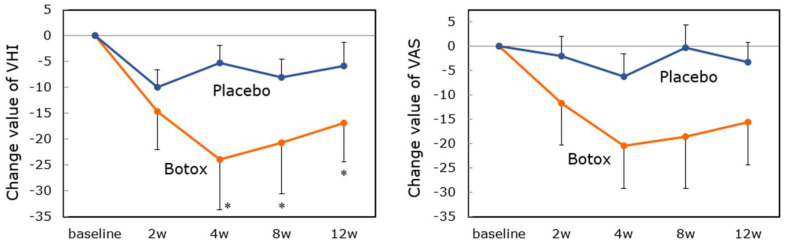
Change values of subjective parameters after Botox/placebo injection (Mean ± SE). The Voice Handicap Index (VHI) and Visual analogue scale (VAS) scores improve over 4−12 weeks with peaks at 4 weeks. The scores at 2 weeks do not indicate sufficient improvements. (*: *p* < 0.05; vs. baseline) This means Botox injection improves the voice subjectively after the 4th week.

**Figure 5 toxins-14-00451-f005:**
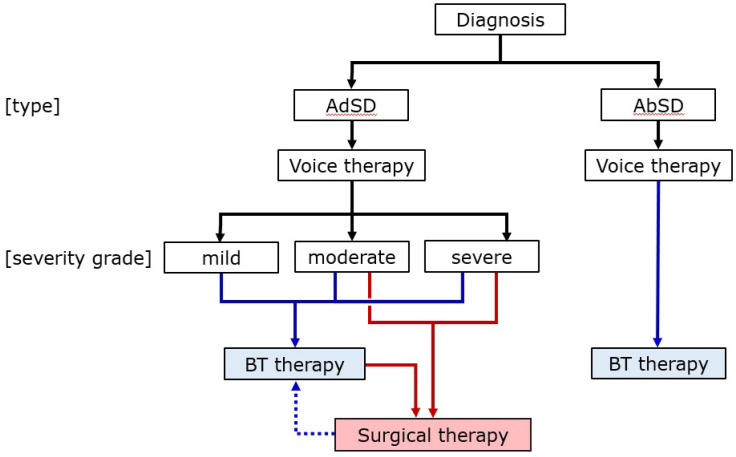
The proposed treatment flow for SD. (Hyodo, et al. Adapted from [27]). Botulinum toxin injection is indicated for AdSD patients regardless of severity. Surgical intervention may be indicated for patients with moderate-to-severe AdSD. If symptoms recur or last postoperatively, botulinum toxin therapy can be performed. For AbSD, botulinum toxin injection is the only treatment method that can be expected to be sufficiently effective. AdSD: adductor spasmodic dysphonia, AbSD: abductor spasmodic dysphonia, BT: botulinum toxin.

**Table 1 toxins-14-00451-t001:** Research achievements and future investigations on SD in Japan.

	Research Achievement	Issues to be Studied
Epidemiology	Nationwide survey (Yamazaki, 2001 [23]; Hyodo, 2015 [24])	Periodical survey
Diagnosis	propose of diagnostic criteria (Hyodo, 2021 [27])	Need for validation
Objective grading of severity	Establishment of mora method (Kumada, 1997 [44])	Apply to other language
Botulinum toxin therapy	Placebo-controlled double-blinded clinical trial (Hyodo, 2021 [46])	Large-scale study by many patients
Thyroplasty type 2	Clinical trial of titanium bridge (Sanuki, 2022 [51])	Expand to worldwide

## Data Availability

The data used to support the findings of this study are included within the article.

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
