# Peer review of "Botulinum Toxin Therapy for Spasmodic Dysphonia in Japan: The History and an Update"

_toxins, 2022, doi:10.3390/toxins14070451_

Round 1

Reviewer 1 Report

Please add the dose of BT for voice treatment. Is there any optimal or standard one that is determined, or is that individually calculated?

In most literature, the botulinum toxin abbreviation is suggested to be used harmonized, please follow that. See BoNTA for example.

Which type of BT was injected? A, or B or? If the most applied one is the BoNTA then please indicate the type correctly.

In Fig 1, the Y-axis is the % of “what”? please make sure that the axis title and units are clear.

In Fig 2, the Y-axis title is the subjective score, please correct the misspelling of the subjective score. In addition, it is important to explain and clarify what is the subjective score in the figure legend, so that the figure can stand alone understandable, without the need to get back to the text.

In Fig 3, please add the title of the Y-axis, and also explain if the negative scores mean reduction or improvement. This needs to be clarified in the figure legend. Please see the above comment. Also, please add what the whiskers are, are those SD for means or SEM (or other)? Please follow these points for figure 4, too.

Please define the abbreviations in Fig legend of Fig 5.

Please add the safety aspects of SD challenges and limitations of this review (it is not a systematic review, so the authors are encouraged to add the type of review and if this can yield bias of ignoring some publications in the literature search).  

It is beneficial to add a summary table to give an overview of what has been done and show the missing points, or gaps in the literature so that the next step of research can close the existing gaps.

Is there any age or sex dependency on the outcome so far? Please elaborate.

Are there any contraindications in this type of therapy? Please elaborate.

Author Response

Thank you for your peer review and valuable comments. We have corrected the manuscript as below.

1) Please add the dose of BT. Is there any optimal or standard one that is determined, or is that individually calculated?

I added a description about the treatment dose of botulinum toxin in page 4.

2) In most literature, the botulinum toxin abbreviation is suggested to be used harmonized, please follow that. See BoNTA for example.

In this article, “botulinum toxin” does not necessarily mean only botulinum toxin A. Thus, abbreviation BoNTA may not be adequate. To avoid confusion, we decided not to use the abbreviation BT and described it as “botulinum toxin” excepting in some figures.

3) Which type of BT was injected? A, or B or? If the most applied one is the BoNTA then please indicate the type correctly.

In Japanese clinical trial, type A toxin (Botox) was used. This is described. (Page 5)

4) In Fig 1, the Y-axis is the % of “what”? please make sure that the axis title and units are clear.

In Fig 1, we added a title of Y-axis; previously received treatments.

5) In Fig 2, the Y-axis title is the subjective score, please correct the misspelling of the subjective score. In addition, it is important to explain and clarify what is the subjective score in the figure legend, so that the figure can stand alone understandable, without the need to get back to the text.

We corrected the misspelling of “subjective”. Also, we added an explanation of the subjective score in the figure legend.

6) In Fig 3, please add the title of the Y-axis, and also explain if the negative scores mean reduction or improvement. This needs to be clarified in the figure legend. Please see the above comment. Also, please add what the whiskers are, are those SD for means or SEM (or other)? Please follow these points for figure 4, too.

In Fig 3 & 4, title of the Y-axis was added and explained that negative value means voice improvement in the figure legends. The whiskers are SEs and “Mean ± SE” was added in the figure legends.

7) Please define the abbreviations in Fig legend of Fig 5.

We defined the abbreviations in Fig5; AdSD, AdSD, and BT.

8) Please add the safety aspects of SD challenges and limitations of this review (it is not a systematic review, so the authors are encouraged to add the type of review and if this can yield bias of ignoring some publications in the literature search).  

This paper is not a systematic review. This was described as a limitation in a section of “7. Limitation and significance of this review” in page 7. Safety of the treatment was added in page 3 (line 132) and page 6 (line 233).

9) It is beneficial to add a summary table to give an overview of wha.t has been done and show the missing points, or gaps in the literature so that the next step of research can close the existing gaps.

We added a research summary table. (Page 8)

10) Is there any age or sex dependency on the outcome so far? Please elaborate.

Differences of the treatment outcome were added in page 6 (line 225-233).

11) Are there any contraindications in this type of therapy? Please elaborate.

Contraindications of botulinum toxin therapy for spasmodic dysphonia was described in section 4.2. (page 4)

Reviewer 2 Report

An interesting review. I suggest some minor corrections which I feel should be of interest for the readers: 

1. Describe data on the diagnostic of spasmodic dysphonia in clinical practice

2. Describe the method (doses, muscles of injection recommended, use of EMG, or other, etc) of botulinum toxin infiltration for this disease.

3. Finally, a description of strengths and limitations of the review should be appropriate.

Author Response

Thank you for your peer review and valuable comments. We have corrected the manuscript as below.

1) Describe data on the diagnostic of spasmodic dysphonia in clinical practice.

We added a section of “3. Diagnosis of SD” in page 3.

2) Describe the method (doses, muscles of injection recommended, use of EMG, or other, etc.) of botulinum toxin infiltration for this disease.

We added a section of “4.2. Procedure of botulinum toxin therapy” in page 4 and described the treatment method including doses, target muscles, and injection approaches.

3) Finally, a description of strengths and limitations of the review should be appropriate.

We added a description of limitations and significance of this review article in page 7.

Round 2

Reviewer 1 Report

The authors have taken into account the reviewers comments and suggestions and accordingly revised the manuscript. Thy have also provided a point by point explanation. There are no more comments.